# Physical Properties of Thermally Crosslinked Fluorinated Polyimide and Its Application to a Liquid Crystal Alignment Layer

**DOI:** 10.3390/polym13223903

**Published:** 2021-11-11

**Authors:** Jong-Soo Ahn, Su Hong Park, Na Yeon Kwon, Min Ju Cho, Sang-Hyon Paek, Dong Hoon Choi

**Affiliations:** 1Department of Chemistry, Research Institute for Natural Science, Korea University, 145 Anam-ro, Seongbuk-gu, Seoul 02841, Korea; jsahnlcd@hyrubbing.com (J.-S.A.); parkshsd@korea.ac.kr (S.H.P.); 2018010906@korea.ac.kr (N.Y.K.); 2Department of Chemical Engineering, Kyung Hee University, 1732, Deogyeong-daero, Giheung-gu, Yongin-si 17104, Korea

**Keywords:** polyimide, crosslinking, elastic modulus, liquid crystal cell, voltage holding ratio, residual direct current voltage

## Abstract

This study demonstrated the use of a thermally crosslinked polyimide (PI) for the liquid crystal (LC) alignment layer of an LC display (LCD) cell. Polyamic acid was prepared using 4,4′-oxydianiline (ODA) and 4,4′-(hexafluoroisopropylidene) diphthalic anhydride (6FDA). The 6FDA−ODA-based polyimide (PI) prepared by the thermal cyclic dehydration of the polyamic acid (PAA) was soluble in various polar solvents. After forming a thin film by mixing trifunctional epoxide [4-(oxiran-2-ylmethoxy)-N,N-bis(oxiran-2-ylmethyl)aniline] with the 6FDA−ODA-based PAA, it was confirmed that thermal curing at −110 °C caused an epoxy ring opening reaction, which could result in the formation of a networked polyimide not soluble in tetrahydrofuran. The crosslinked PI film showed a higher rigidity than the neat PI films, as measured by the elastic modulus. Furthermore, based on a dynamic mechanical analysis of the neat PI and crosslinked PI films, the glass transition temperatures (T_g_s) were 217 and 339 °C, respectively, which provided further evidence of the formation of crosslinking by the addition of the epoxy reagent. After mechanical rubbing using these two PI films, an LC cell was fabricated using an anisotropic PI film as an LC alignment film. LC cells with crosslinked PI layers showed a high voltage holding ratio and low residual direct current voltage. This suggests that the crosslinked PI has good potential for use as an LC alignment layer material in advanced LCD technologies that require high performance and reliability.

## 1. Introduction

The liquid crystal (LC) alignment layer (AL) is an essential component of an LC display (LCD). This causes an alignment of anisotropic LC molecules to respond continuously and uniformly according to an applied voltage. To ensure a uniform LC alignment, LC AL materials must satisfy several important characteristics as follows. (1) The ion-free AL material must be quite pure. (2) It must have the ability to form a robust film with excellent dimensional stability, excellent chemical/thermal stability, good compatibility with LC molecules, excellent optical transparency, and a high surface hardness to withstand the mechanical rubbing process [1,2,3].

Among the various polymer candidates, polyimide (PI) is known as an excellent candidate that can satisfy the various requirements mentioned previously [4,5,6,7,8]. Thus, it is used as an AL material in various LCD modes. Among the unique characteristics of various LC cells, the high voltage holding ratio (VHR) and residual direct current voltage (RDC) are known as crucial factors in achieving a high-quality, full-color LCD. In general, the VHR and RDC values of LC cells are closely related to brightness, contrast, and image sticking issues, which depend on the material properties of the PI AL [9,10,11]. To achieve high VHR and low RDC values in LC cells, LC AL materials with high purity, high chemical resistance, and high electrical resistance values are required. Therefore, the structural design of the PI AL material should be considered to satisfy these requirements.

PI is quite a stiff material in terms of its mechanical properties and is known as a polymer that can be used to manufacture free-standing films or coat various substrates, either as PI itself or using polyamic acid (PAA). In addition, after the imidization reaction in PAA, PI often becomes remarkably resistant to organic solvents. PIs that are soluble in organic solvents have also been reported, depending on the structures of the dianhydrides and diamines used for the synthesis [12,13,14,15,16]. Among the PIs with various structures, soluble 6FDA−ODA is readily obtained from the thermal or chemical imidization of PAA synthesized with 4,4′-oxydianiline (ODA) and 4,4′-(hexafluoroisopropylidene) diphthalicanhydride (6FDA) [17]. Unlike the conventional insoluble PI, 6FDA−ODA has two CF_3_ groups in the 6FDA unit, providing bulky space in the polymer chains. It has a relatively low mechanical modulus and a low glass transition temperature (T_g_) because the bulky space moieties disturb the dense packing between PI chains, providing a relatively large free volume.

However, although 6FDA−ODA-based PI has advantages in terms of processability, as mentioned previously, controlling the mechanical properties and solubility in organic solvents while maintaining the structural properties of 6FDA−ODA-based PI is often necessary. If a crosslinking reaction between the PAA chains is induced prior to imidization, the packing between the two chains would be expected to be maintained, and the solubility of the final 6FDA−ODA could be controlled. Therefore, if a method for converting soluble PI to insoluble PI was developed, it could be very useful for limited special applications of PI [18]. An epoxy compound could be selected as an effective agent for the crosslinking between PAA chains, along with the reaction between diamines or diols, which is well known. It is advantageous to induce a temperature much lower than the ring-closed imidization reaction temperature [19,20].

Meanwhile, when preparing an LC AL for manufacturing an LC cell, a high surface hardness for the AL material against mechanical friction is required. In addition to an improvement in the anisotropic stability of the PI chains after mechanical rubbing and uniformity of the LC alignment in the microgrooves of the rubbed surface, a harder alignment material is generally preferred [21,22]. Therefore, in general, when 6FDA−ODA-based PI is applied as an LC AL, it is expected that a neat PI layer and crosslinked PI layer will show the above differences in their properties.

In this study, 6FDA−ODA-based PI films and crosslinked PI films were easily prepared via the thermal imidization of 6FDA−ODA PAA films with and without [4-(oxiran-2-ylmethoxy)-N,N-bis(oxiran-2-ylmethyl) aniline] (EP) as a crosslinkable epoxy reagent. The 6FDA−ODA PI film was well soluble in THF, whereas the crosslinked 6FDA−ODA PI film was insoluble in THF. A tensile test and dynamic mechanical analysis (DMA) of each of the above two PI films showed that the crosslinked PI film had a relatively high elastic modulus and glass transition temperature (T_g_), as expected. To prepare the LC AL, 6FDA−ODA-based neat and crosslinked PI films were coated onto indium tin oxide (ITO) glass substrates, and surface mechanical rubbing was performed. When these two types of PI films were used as LC ALs, the VHR and RDC of the crosslinked PI-based LC cell were relatively good due to their unique physical properties. Therefore, the strategy of introducing crosslinking to soluble PI suggests the possibility of its application to advanced LC AL materials, which require high performance and reliable LCDs.

## 2. Materials and Methods

### 2.1. Materials

The polyamic acid solution (4.5 wt% in NMP) used for the 6FDA-ODA was supplied by NCK Co., Ltd. (Pyeongtaek, Korea). The [4-(oxiran-2-ylmethoxy)-N,N-bis(oxiran-2-ylmethyl)aniline] (EP) epoxy was purchased from Sigma-Aldrich. The MLC-2041 (IPS mode LC) used for the actual mass production of LCDs was purchased from Merck and used for the LC cell fabrication.

### 2.2. Preparation of Neat 6FDA−ODA PI and Its Crosslinked PI Films

A thin film of PI was applied to the substrate by spin coating using a PAA solution with or without epoxy (EP). According to the findings of a preliminary study, a concentration of 5 wt% EP mixed with the PAA film was chosen (Appendix A). In addition, a thick film was prepared by bar coating at room temperature using the same high-concentration PI solution. The prepared PAA film was dried in a high-temperature vacuum drying oven (110 °C for 2 h, 120 °C for 1 h). Then, the final 6FDA−ODA PI film was obtained by thermal imidization in a vacuum oven at 230 °C for 30 min. In addition, a PAA film containing a small amount of EP in PAA was dried using the same process as the neat PAA film to induce a crosslinking reaction. By heat-treating the crosslinked PAA film at 230 °C, most of the repeating groups of PAA that did not participate in the crosslinking were imidized.

### 2.3. Instrumentation

Thermogravimetric analysis (TGA) with a Mettler STARe was used to determine the decomposition temperatures (T_d_s) of the PAA and PIs. The T_d_ of the polymer was defined as the temperature at which a 5% weight loss occurred. UV-vis absorption spectra were recorded using an Agilent 8453 UV–vis spectrophotometer. Fourier transform infrared (FT-IR) analysis was conducted using a Bio-Rad FTS-60A infrared spectrometer in attenuated total reflection mode. A reflection cell with an incidence angle of 60° was used. An average of 128 scans were collected for each film sample over the range of 400–4000 cm^−1^. The resolution was 4 cm^−1^. The dielectric constants of the PI films were measured, using an Agilent LCR meter (ASTM D150), at room temperature while changing the frequency from 10 Hz to 10^6^ Hz using the scanning mode. Atomic force microscopy (AFM) was conducted in the contact mode with silicon tips using an AFM 5100N from the Hitachi High-Tech Sci. Co. Tensile tests of the PI films were performed at a tensile speed of 10 mm min^−1^ on dog-bone specimens (50 mm × 5 mm × 40 μm) using a Shimadzu AGS-100G (Shimadzu Corporation, Kyoto, Japan). DMA analysis of the thick PI films was performed using a DMA Q800 (TA Co., Detroit, MI, USA)at 1.0 Hz and a heating rate of 5 °C∙min^−1^ from room temperature to 400 °C.

### 2.4. Characterization of AL

The alignment treatment was performed by mechanically rubbing the PI films. Using an automatic rubbing machine that could be controlled up to a 0.01 mm pile impression, the PI alignment film was rubbed once with a Hyperflex Sheath-Core type rubbing cloth (HY-7018, Hyperflex Co., Ltd., Gumi, Korea) (rubbing roller diameter: 440φ, rubbing roller rpm: 1000 rpm, pile impression: 0.3 mm, stage speed: 20 mm/s). The optical anisotropy of the rubbed LC alignment layer was measured using LayScan (λ = 532 nm, 50 mW, angle of incidence = 50°, Moritex Co., Ltd. (Asaka, Japan), LYS-LH3040). When p-polarized light with a vibration direction parallel to the incident surface of the alignment layer, having optical anisotropy, or s-polarized light with a vibration direction perpendicular to the incident surface is scanned, light containing different polarization components is reflected and its intensity is measured. The surface conditions of the PI films were estimated by measuring the surface roughness of each polymer film in a 100 μm^2^ area using AFM.

### 2.5. Fabrication and Characterization of LC Cell

A 100 nm thick polymer film was prepared by spin-coating a PAA solution on a glass substrate with an ITO electrode, after which the PAA thin film was dried at 80 °C for 1 min, followed by heat treatment at 230 °C for 30 min to prepare the PI layer. A dry method was used to spread a spacer with a diameter of 4.0 μm on one of the two ITO/glass substrates with a rubbing PI alignment film. Subsequently, an IPS cell (anti-parallel) with a cell gap of 4.0 μm was fabricated by bonding the two substrates so that the rubbing directions were 180° to each other. In particular, in the manufacture of the LC cell for the measurement of electrical properties, the two substrates were bonded to ensure that the pattern shapes of the upper and lower plates were matched. Then, when the assembled cell was pressed, it was treated in a clean oven at 80 °C for 15 min and 150 °C for 90 min to heat cure the epoxy sealant. The measurement of the cell gap for the empty cell using an OPTIPRO-micro measuring instrument (SHINTECH) showed that the cell gap was in the range of 3.8–4.2 μm in all cases. After injecting MLC-2041 LC into the empty cell at room temperature and atmospheric pressure, the LC inlet was sealed with an end sealant (XN-1500T from Mitsui Chemicals), and the LC part was masked to protect the AL and LC from UV light. The production of the LC cell was completed by final curing under UV light irradiation. The VHR of the LC cells was measured using a VHR: 6254C from the Toyo Corporation. After a constant voltage was applied to the LC cell, the degree of voltage maintained by the cell compared to the applied voltage during the period of no voltage application was called the VHR. The VHR of the LC cell was measured at 23 °C and 60 °C with a pulse width of 64 µs, a frequency of 60 Hz, pulse widths of 167 ms and 1667 ms, and data voltage of 4.0 V. The RDC values were also measured using Toyo 6254C.

## 3. Results and Discussion

### 3.1. Preparation and Characterization of PI Films

The 6FDA−ODA-based PAA used in this study had an average molecular weight (M_n_) of 14.6 kDa (PDI = 3.16) and showed good film-forming properties. Thin and thick films were prepared using the spin-coating and bar coating methods, respectively, under ambient conditions. As shown in Figure 1, the corresponding PI was prepared by imidization at 230 °C with the 6FDA−ODA-based PAA. When preparing a crosslinked PAA film bearing 5 wt% of EP, the film was cured at 110 °C for 2 h and 120 °C for 1 h to dry the PAA film and induce a crosslinking reaction between the PAA chains. Then, the cured film was heated to 230 °C for the final imidization of the repeating groups that did not participate in the crosslinking between the PAA chains. To observe the change in the structure of the 6FDA−ODA-based PAA during imidization, it was necessary to observe the changes in the amide and imide bands in the FT-IR spectrum. To monitor the crosslinking and imidization processes, the FT-IR spectra of neat PAA films and PAA films with 5 wt% EP heat-treated at different temperatures, were investigated.

In the spectrum of the neat PAA film, as shown in Appendix A, an N–H stretching band at 2900–3200 cm^−1^, a carbonyl stretching band of carboxylic acid at 1722 cm^−1^, and a symmetric carboxylate stretching band at 1409 cm^−1^ can be observed. In addition, carbonyl stretching bands of the amide I and amide II modes were observed at 1659 cm^−1^ and 1543 cm^−1^, respectively. As the temperature of the sample gradually increased, a change in the spectrum was observed. In particular, it was possible to observe a structural change in the PAA to the corresponding PI after raising the temperature of the sample to 200 °C or more. The absorption bands around 1543 cm^−1^ (amide II) and 1659 cm^−1^ (amide I) disappeared as the imidization progressed. Simultaneously, it was observed that carbonyl stretching bands (imide I) appeared at 1785 cm^−1^ (asymmetric stretching) and 1726 cm^−1^ (symmetric stretching). Additionally, a typical C-N stretching (imide II) vibration band was observed at approximately 1377 cm^−1^. The conversion reaction from PAA to PI was confirmed by the characteristic bands mentioned above. On the other hand, the FT-IR spectrum of the PAA film containing 5 wt% EP showed a spectral change similar to that of the pure PAA film after heat treatment. No evidence of ring-opening of the epoxy ring was found because the amount of EP was relatively small; thus, almost no variation in the characteristic peaks was observed as a result of the crosslinking. Therefore, in order to obtain evidence of the occurrence of the crosslinking reaction in the samples prepared by mixing 6FDA−ODA-based PAA and 5 wt% EP and to observe whether curing occurs at low temperature (50 or 70 °C) upon increasing EP concentration to 30 wt%, we performed FT-IR spectroscopy. We observed that the IR spectrum of the PAA sample mixed with 30 wt% EP was very similar to that of the neat PAA sample, except the slight differences in the peaks detected at around 1038 cm^−1^ for the neat EP, as shown in Appendix A. On the other hand, the 898 cm^−1^ band of the neat PAA sample showed a significant overlap with the 906 cm^−1^ band of the neat epoxy compound, as shown in the FT-IR spectra of PAA with 30 wt% EP. We expected that the epoxy band would disappear when crosslinking proceeded, and only the neat PAA band would remain.

Assuming this, we calculated the degree of conversion based on the change in absorbance at 906 cm^−1^ according to the curing time by fixing the curing temperature. In addition, the 1070 cm^−1^ band of neat PAA, which did not exhibit changes during crosslinking, was determined as a reference band. We assumed that at 100% conversion, the 906 cm^−1^ band of the epoxy compound would disappear completely. Therefore, the 906 and 1070 cm^−1^ absorbance ratio for neat PAA was used as the ratio representing 100% conversion. Based on the FT-IR results as shown in Appendix A, the degree of conversion according to the number of reacted epoxy groups was determined using the following Equation (1):(1)$$\mathrm{Degree}\;\mathrm{of}\;\mathrm{conversion}\;\left(\%\right)=100\;\left[\;1-\frac{{\left(A_{906}/A_{1070}\right)}_{t}}{{\left(A_{906}/A_{1070}\right)}_{0}}\right]$$
where *A*_906_ and *A*_1070_ represents the absorption intensity at 906 and 1070 cm^−^^1^, respectively, and the subscript t denotes the reaction time. In this experiment, we observed that increasing the curing temperature from 50 to 70 °C increased the curing rate and shortened the reaction completion time as shown in Appendix A. The approximate rate constants of the crosslinking reaction in an initial stage of heat treatment were determined to be 0.29 and 0.76 min^−^^1^ for the reaction at 50 and 70 °C, respectively. The above results confirmed that a crosslinked structure between PAA and EP was formed; on this basis, it was expected that a crosslinked structure could similarly be formed, even in the case of the PAA sample mixed with 5 wt% EP. (Appendix A). In this study, to prepare the final crosslinked PI, drying at 110 °C (step 1) and heat treatment at 230 °C for imidization (step 2) were performed. The above-mentioned experimental results confirm that crosslinking occurred effectively during the drying process of the PAA blend film.

### 3.2. Thermal Properties of Neat PI and Crosslinked PI

TGA thermograms of the 6FDA−ODA-based PAA and PI films with and without the EP crosslinker are shown in Figure 1. The films were measured in the temperature range 50–900 °C in nitrogen. When the PAA film was dried in a vacuum oven at 100 °C for 2 h and 120 °C for 1 h, a two-stage weight loss trend was observed (curve A). The PAA was imidized at a temperature above approximately 150 °C and showed an initial weight loss, and the imidization of the PAA was completed at approximately 300 °C. Moreover, this sample showed no weight loss from 300 °C to 550 °C, indicating that the imidization was complete at 300 °C. This indicated that the prepared 6FDA−ODA-based PI was thermally stable up to 550 °C. In the case of the PI subjected to full imidization at 230 °C, as shown in curve B, almost no weight loss occurred up to 550 °C. Finally, a significant weight loss occurred again at approximately 550 °C as a result of the pyrolysis reaction of the PI chains. On the other hand, the PAA film containing EP (5 wt%), dried at 100 °C for 2 h and 120 °C for 1 h in a vacuum oven, showed a thermogram (curve A’) similar to that of the above-mentioned neat PAA film. While the temperature of the sample was being raised to 300 °C, a crosslinking reaction occurred in the range 100–120 °C, and the imidization of the repeating units that did not participate in the crosslinking reaction occurred at 230 °C. Therefore, it is believed that a small weight loss occurred as a result of the decomposition of the crosslinking bridge and some water removal from residual imidization above 300 °C. In the case of the PAA sample containing EP annealed at 230 °C for 1 h (curve B’), a small weight loss was observed from 300 °C to 550 °C due to the same thermal decomposition of the crosslinking bridges, as shown in curve A’.

### 3.3. Solvent Resistance of Neat 6FDA−ODA PI and Crosslinked PI

In the 6FDA−ODA PI structure, the presence of 1,1,1,3,3,3-hexafluoro-2,2-dimethylpropane units as bulky spacers in the repeating unit induces steric hindrance between the polymer chains. Therefore, the PI synthesized using 6FDA is soluble in organic solvents due to the interchain steric hindrance. Therefore, the polymer chain packing of 6FDA−ODA is poor, and an amorphous region can easily form. Therefore, it is soluble in common polar solvents such as tetrahydrofuran (THF), dimethylformamide (DMF), and N-methyl-2-pyrrolidone (NMP). Meanwhile, when thermal crosslinking was induced by adding 5 wt% EP to the PAA solution, the solvent resistance of the 6FDA−ODA-based PI was significantly improved.

Figure 2a shows the results of the UV-vis absorption spectroscopy study of the 6FDA-ODA PI films, demonstrating their solubility in THF. The absorption spectra recorded after thermal imidization (T_imidization_ = 230 °C) of the 6FDA-ODA-based PAA films and rinsing with THF show a significant decrease in absorbance in the wavelength range 250–300 nm. This indicates that the existing PI was partially dissolved in THF. Meanwhile, when the PAA film, to which 5 wt% EP was added, was heat-treated at 230 °C, almost no decrease in absorbance was observed in the above wavelength range, even after rinsing with THF (Figure 2b). We also performed solubility tests of the crosslinked PI in DMF and NMP solvents and the results are in the Appendix A. As a result, it can be expected that after adding a small amount of EP to PAA, the solubility of the final PI is completely changed after thermal treatment because a crosslinking reaction between the PI chains occurs between the PAA chains.

### 3.4. Mechanical Properties of Neat PI and Crosslinked PI Free-Standing Thick Films

Representative tensile stress–strain (s–s) curves and specimens of the neat PI and crosslinked PI free-standing films are shown in Figure 3a. The tensile strength and elastic modulus (Young’s modulus) of the neat 6FDA−ODA-based PI film were determined to be 57.4 MPa and 1.85 GPa, respectively, while the crosslinked PI film exhibited values of 72.9 MPa and 2.51 GPa, respectively. Therefore, the crosslinked PI film showed higher stiffness and ultimate stress values at break than the neat PI film. The tensile strain at break of the crosslinked PI was 0.040, smaller than that (ε = 0.049) of the neat PI film without plastic deformation, as expected. This different mechanical behavior is attributed to the formation of crosslinks between the PI chains.

Figure 3b shows the DMA results for the 6FDA−ODA-based neat PI and crosslinked PI free-standing thick films. The glass transition temperature (T_g_) of the neat PI film was observed to be 217 °C, with a tan *δ* value of 0.34. Imidization of PAA is expected to be induced at temperatures below the T_g_, leading to high levels of chain packing at temperatures above 230 °C. The T_g_ of the crosslinked PI (339 °C) was much higher than that of the neat PI, and the tan *δ* value was approximately 0.26, which is smaller than that of the neat PI film. This indicated that the crosslinked PI film had a more solid-like viscoelastic behavior.

### 3.5. Dielectric Properties of Neat PI and Crosslinked PI Free-Standing Thick Films

Next, the capacitance values of the 6FDA−ODA PI and crosslinked PI free-standing films were measured in order to compare their dielectric properties. It is generally known that the dielectric properties of a polymer such as 6FDA−ODA PI change when the polymer repeating unit contains fluorine, which is attributed to a relatively low polarizability and large free volume [23,24]. In general, when a PI film is used as an LC AL, the dielectric constant of PI affects the degree of accumulation of static charges formed on the surface after the rubbing process and also affects the time the residual charges remain on the surface. It is known that the low charge accumulation of a PI film made of PAA can minimize the image sticking problem in LCDs. The effect of crosslinking on the dielectric properties is clearly observed in Appendix A. When the dielectric constant and dielectric loss factor were specifically analyzed according to the frequency of the PI film sample, it was observed that the crosslinked PI film had a lower dielectric constant (ε = 3.69 at 10 kHz) than the neat PI film (ε = 4.02 at 10 kHz). In addition, the crosslinked PI film exhibited very stable dielectric properties at this frequency compared to the neat PI film. This phenomenon is due to the synergistic effect of the large free volume and loosening of the molecular packing due to the formation of partial crosslinking bridges between the PI chains. Finally, it can be inferred that the formation of a cross-linked structure between the PI chains can change the internal structure of the film, leading to a change in the dielectric properties of the film.

### 3.6. Surface Morphologies of Neat PI and Crosslinked PI Films: Effect of Mechanical Rubbing

The function and stability of the 6FDA−ODA PI film for application as an LC AL are closely related to the surface morphology of the PI film in contact with the LC and the stability of the rubbed PI film. Therefore, the surface morphological properties of the rubbed 6FDA−ODA PI film were investigated using AFM. For the neat PI films, the root mean square surface roughness (R_q_) slightly increased from 0.302 nm before rubbing to 0.460 nm after rubbing, as shown in Figure 4a,b. Conversely, the surface morphology of the crosslinked PI film was not significantly affected by rubbing (Figure 4c,d). This means that the strong friction between the rubbing cloth and the PI film did not significantly affect the morphological stability of the surface microgrooves on the crosslinked PI film. Therefore, the crosslinked PI film maintained its surface uniformity and the morphological robustness of the microgrooves even after rubbing because the surface dent occurred within the elastic limit of the PI film during rubbing. For this reason, when the crosslinked PI film is used as an LC AL, it is expected to have high reliability for LC alignment in the cells.

### 3.7. Rubbing Effects on Anisotropy of Neat PI and Crosslinked PI AL

Mechanical rubbing of a PI AL with a rubbing cloth caused changes in the surface morphology of the AL. First, the top layer of the AL surface in contact with the rubbing cloth fibers during the rubbing process was removed by friction action; thus, microgrooves were formed on the AL surface in the rubbing direction. This resulted in a rougher AL surface and a larger surface area. Second, the PI chains near the AL surface were preferentially oriented in the rubbing direction by the rubbing force. These two rubbing effects consequently generated anisotropy in the PI AL. The degree of this rubbing-induced anisotropy of the AL depended on the chemical and mechanical properties of the AL material, as well as the rubbing conditions, including the rubbing force, rubbing length, speed of the rubbing roller, and rubbing temperature. Figure 5 shows the changes in the intensity of the linearly polarized light passing through the AL with respect to the AL’s in-plane angle for both the 6FDA−ODA PI film (a) and its crosslinked PI film (b) before and after rubbing. The maximum intensities in Figure 5 are shown for angles of 45° and −135°, which correspond to the rubbing direction; on the other hand, 135° and −45° corresponded to the perpendicular direction and showed the minimum intensities. The average differences between the maximum and minimum intensities indicated the relative degree of rubbing-induced anisotropy; that is, a larger difference in intensity indicates a larger anisotropy in the AL. The asymmetric polar diagram might be due to the inclination of polyimide chains in the vertical direction or variation of the light incident spot during measurement.

Significant anisotropy was generated in the AL by rubbing, and its degree is obviously larger for the 6FDA−ODA AL (Figure 5a). This indicates a more significant change in the PI chain orientation as a result of rubbing, which matches the result in Figure 4 showing the rougher surface of the rubbed 6FDA−ODA PI film. This higher rubbing effect for the 6FDA-ODA PI AL is due to its inferior mechanical properties, such as its modulus and tensile strength, compared to the crosslinked PI film.

### 3.8. Properties of LC Cell: VHR and RDC

The VHR is a measure of an LCD’s ability to maintain the voltage applied to its pixels during a frame time interval for active addressing; thus, a higher VHR is required to produce high-quality images [25,26,27]. It is known that the VHR is reduced by ionic charges and impurities (including radicals) present in the LC material/layer. Moreover, these ionic charges and/or radicals in the LC layer can be absorbed (or accumulated) on the AL surface under repetitive application of an external DC offset voltage, which leads to the generation of an RDC [28,29,30]. Thus, the VHR is related to the RDC; in general, the VHR tends to decrease with an increasing RDC [30,31]. The RDC is one of the major causes of the image sticking phenomenon, therefore, the minimization of the RDC is also desired for better display performance [10,29].

Generation of an RDC is based on the adsorption and desorption of ionic charges/components at the interface between the LCs and PI AL. Thus, the RDC and VHR characteristics of an LC cell depend on the chemical nature of the AL PI, as well as on the intrinsic properties and purity of the LC material, which determine the interactions of the LCs with the PI molecules [8,28,31]. Figure 6a shows the VHR values of LC cells with the 6FDA−ODA-based neat PI and crosslinked PI ALs. The VHRs were determined by varying the voltage holding time (167 ms and 1667 ms) and temperature (23 °C and 60 °C).

In general, the VHR decreased considerably as the temperature increased. This was due to the mobility of the ionic impurities increasing at an elevated temperature [8,28]. The VHRs of the LC cell with the crosslinked PI AL were higher under all measurement conditions. A higher VHR value indicates that the desired pixel image could be less deteriorated during a time interval for LCD addressing. Furthermore, the VHRs for the crosslinked PI AL decreased less as the temperature and voltage holding time increased, indicating better stability in the VHR value. The crosslinked PI AL also induced a better RDC characteristic in the LC cell than the 6FDA−ODA PI; the RDC values for the former AL were smaller, as shown in Figure 6b. Rubbing a PI AL with a rubbing cloth is a process involving contact between their surfaces, therefore, triboelectric charging (or effect) occurs on the surface of the rubbed PI AL as a result of the contact. The electrical (or electrostatic) charges can, therefore, remain on the rubbed PI surface, sometimes for a relatively long time. These charges can increase ionic impurities in the LC material in contact with the rubbed AL and/or act as spots to generate an RDC. As previously mentioned, the results shown in Figure 4 and Figure 5 indicate that the rubbing effect was greater for the 6FDA−ODA PI film, and thus, a rougher and larger surface area was generated by rubbing. Therefore, it is likely that the triboelectric charging was also greater for the rubbed 6FDA−ODA PI AL, which was responsible for the larger RDC and, subsequently, the lower VHR compared to the rubbed crosslinked PI AL. The results in Figure 6a,b clearly show that the LC AL of the crosslinked PI induced better VHR and RDC characteristics in the LC cell, due to better stability of the LCs, as well as less changes in its surface during rubbing. These results suggest that the 6FDA−ODA-based crosslinked PI that incorporated a small amount of crosslinking provided a reliable AL for better LCD performance compared to the neat PI AL.

### 3.9. Chemical Stability of Neat PI and Crosslinked PI Films

The LC AL is a polymer material in direct contact with the LC in the cell. Therefore, the most important property of the AL material is that it must not be swollen or dissolved by the LC used. In particular, the stability of the surface anisotropy formed after mechanical rubbing, which is the most important factor for LC alignment, is also a very important characteristic. To investigate the stability of the anisotropy of the rubbed PI films, the retention of the surface anisotropy of the PI AL was measured after dropping the LC on the PI AL and leaving it at room temperature or at 60 °C. Figure 7 shows the relative change in the anisotropy of the rubbed AL with respect to the wetting treatment with LC (MLC−2041) at 60 °C. As expected, the anisotropy of the rubbed AL decreased with increasing time for the LC treatment. However, the degree of this decrease in the anisotropy was much smaller for the crosslinked PI AL, indicating that the rubbing-induced orientation of the crosslinked PI AL was less deteriorated, and the crosslinked PI was more stable in the LC material. This better stability of the crosslinked PI AL allowed less damage to the LCs, and, therefore, reduced the possibility of increasing ionic impurities in the LC layer by contamination. It is expected that the dimensions of the microgrooves formed on the surface would be maintained due to the stability of the crosslinked PI surface during the LC contact. In brief, it is thought that the high VHR and low RDC values of the corresponding LC cells using the crosslinked PI AL observed earlier were mainly a result of high anisotropy retention.

## 4. Conclusions

This study investigated a crosslinking strategy to compensate for the various shortcomings of soluble 6FDA−ODA PI. A small amount of epoxy compound was introduced as a crosslinking agent in PAA to prepare a crosslinked PI. The crosslinked PI film was found to be superior to the neat PI film in terms of its solvent resistance and thermal stability, and its elastic modulus and tensile strength. The performances of LC ALs that used the 6FDA−ODA-based PI and its crosslinked PI in LC cells were compared. In summary, the reliability of LC cells with the crosslinked PI AL was superior, which was confirmed by evaluating the VHR and RDC values of the LC cell. The material development strategy using crosslinked PI presented in this study was able to improve solvent and heat resistance while maintaining the intrinsic physical properties of PI by easily fabricating a thin film using PI with excellent processability, and then post-processing it. Therefore, the possibility of using crosslinked PI as an LC alignment layer material was unambiguously demonstrated.

## Data Availability

Not applicable.

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
