# Peer review of "Physical Properties of Thermally Crosslinked Fluorinated Polyimide and Its Application to a Liquid Crystal Alignment Layer"

_polymers, 2021, doi:10.3390/polym13223903_

Round 1
Reviewer 1 Report
Paper by Dr. Dong Hoon Choi et al “Physical Properties of Thermally Crosslinked Fluorinated Polyimide and its Application to Liquid Crystal Alignment Layer” describes preparation and study of the alignment properties of new polyimides. Obtained PIs are promising for the application in LC alignment technologies. This paper can be published after the minor revision.
- Lines 21 and 270. It makes no sense to present data on glass transition temperatures with an accuracy of tenths of a degree. What is the accuracy of its determination?
- Section 3.7. Some details of the optical anisotropy determination are lost. For example, wavelength of the light and the origin of the polar diagrams asymmetry in Fig. 5 and Fig. 7.
- The advantages of the new PIs in respect to already known and used in technology should be analyzed and discussed.
Reviewer 2 Report
The article "Physical Properties of Thermally Crosslinked Fluorinated Polyimide and its Application to Liquid Crystal Alignment Layer" by Jong-so Ahn, Su Hong Park, Na Yeon Kwon, Min Ju Cho, Sang-Hyon Paek, and Dong Hoon Choi addresses the technological problem of compatibility, adjustability and solvent resistance of a polyamide used in the construction of LCD cells. The solution is based on the cross-reticulation of the aforementioned polyamide with a trifunctional epoxid.
The article is of high level, of great applicative interest and widely developed for this technological field. In addition, it is fully adequate to the Polymers journal and will be of undoubted interest to its readers.
However, there are some crucial points that authors should further develop:
- It is evident that the molecular weight of the polyamide is not particularly high, such as to ensure its good workability and solubility. It is not clear, however, why the crosslinking reaction was conducted with a 5% of epoxide. Were preliminary studies conducted that led to such a choice? Please explain.
- crosslinking kinetics studies should be conducted. (via calorimetric or IR).
Also, some minor considerations:
- How was the decomposition temperature determined? Is it the temperature at which the sample loses 5 (or 10) % of the original mass, as often reported in the literature? Is it the inflection point of the degradation curve? The authors add this information in the text.
- To make the reading of Figure 1 more immediate, it would be appropriate to rename the curves with symbolism that immediately accounts for the similarity of the treatment. For example, the pair curve (i) and curve (iii) with A and A' and, similarly, curve (ii) and curve(iv) with B and B'.
Reviewer 3 Report
This study demonstrated the use of a thermally cross-linked polyimide (PI) for the liquid crystal (LC) alignment layer of a LC display (LCD) cell. The investigations would be interesting for researcher working in polymer and organic electronics field. The paper could be accepted after the revision.
*Authors of the manuscript tested solubility of the cross-linked films in THF? Other solvents should be also tested.
*Soxhlet extraction method is very useful for demonstration of insolubility of cross-linked material and also amount of uncross-linked left components in the composition. This experiment should be done for the polymers.
*The authors declare that they found glass transition of the cross-linked material at high temperature. Usually well cross-linked materials have not Tg, because macromolecules and even their fragments can not move in the cross-linked material. Maybe the macromolecules are just slightly cross-linked ?
*DSC measurements should be used to demonstrate the Tg.
* The cross-linked films were preliminary tested as layer of a LC display (LCD) cell. Maybe the authors should present more details for the devices. Could properties of the devices compared with those using similar commercial materials for the function in these LC display (LCD) cells ?
Round 2
Reviewer 2 Report
The authors only partially responded to the criticisms.
A study of crosslinking kinetics is completely missing, which is highly recommended for this type of work.
Reviewer 3 Report
After the revision this paper is recommended for publication.
Author Response
Response: The language of the manuscript has been improved after availing the proofreading services of a professional editing company. [https://www.editage.co.kr]
Round 3
Reviewer 2 Report
I appreciated that the authors accepted my suggestion and included kinetic studies, even if with great experimental difficulties.
Therefore, I would have recommended its publication. The problem that the extensive explanation provided in the "Response Letter" is not reported in the text of the article (limiting itself to a brief description at lines 205-210) nor in Supplementary Material, where the figures S3-S5 do not correspond to those mentioned in the article. Was there any file upload error? In addition, it would be appropriate to detail more kinetic studies, not so much to the reviewer, but to the readers.
Round 4
Reviewer 2 Report
Please, pay many attention in supplemetary file. The figure S3-S6 cited in the article do not match with those reported in the supplemetary materials file.